# An Explore-then-Commit Algorithm for Submodular Maximization Under Full-bandit Feedback

**Guanyu Nie**[1]    **Mridul Agarwal**[2]    **Abhishek Kumar Umrawal**[2]    **Vaneet Aggarwal**[2]    **Christopher John Quinn**[1]

[1]Computer Science Department, Iowa State University, Ames, Iowa, USA
[2]Purdue University, West Lafayette, Indiana, USA

## Abstract

We investigate the problem of combinatorial multi-armed bandits with stochastic submodular (in expectation) rewards and full-bandit feedback, where no extra information other than the reward of selected action at each time step $t$ is observed. We propose a simple algorithm, Explore-Then-Commit Greedy (ETCG) and prove that it achieves a $(1 - 1/e)$-regret upper bound of $\mathcal{O}(n^{\frac{1}{3}} k^{\frac{4}{3}} T^{\frac{2}{3}} \log(T)^{\frac{1}{2}})$ for a horizon $T$, number of base elements $n$, and cardinality constraint $k$. We also show in experiments with synthetic and real-world data that the ETCG empirically outperforms other full-bandit methods.

## 1 INTRODUCTION

The stochastic multi-armed bandit (MAB) problem was first introduced by Robbins [1952]. It formalizes challenging sequential decision problems faced by many organizations, including inventory selection, scheduling, work assignments and team formation, multi-market ad campaigns, product recommendation, crowd-sourcing, and investing. The decision maker selects an arm and observes reward that comes from an unknown distribution at each round. The goal of the decision maker is to maximize expected cumulative reward over all rounds. The solution to classical MAB problem demonstrates the trade-off between *exploration* and *exploitation*: should the agent try the arm that has not been tried many times so far (exploration) or should stick with the arm that performed well based on previous observations (exploitation)?

The combinatorial multi-armed bandit (CMAB) problem is an extension of the MAB problem. In this setting, the decision maker selects a *super arm* composed of *base arms* at each round, and observes a reward corresponding to the selected super arm. If the decision maker only learns the aggregated reward for the selected super arm, that feedback is referred to as *full-bandit*. Otherwise, if the decision maker learns additional information (e.g., individual rewards of the base arms), the feedback is referred to as *semi-bandit*. Furthermore, there are two common formalizations depending on the assumed nature of environments: the *stochastic* setting and the *adversarial* setting.

In the adversarial setting, the reward sequence is generated by an unrestricted adversary, potentially based on the history of decision maker's actions [Auer et al., 2003]. In the stochastic environment, the reward of each arm is drawn independently from a fixed distribution [Auer et al., 2002]. For many bandit problems, the stochastic setting is a special case of the adversarial setting. For those problems, algorithms designed for the adversarial setting maintain the theoretical performance guarantees when applied to problems in the stochastic setting, though typically they empirically underperform algorithms specifically designed for the stochastic setting [Lattimore and Szepesvári, 2020]. Moreover, the strategies designed for the stochastic setting may have simpler designs and be computationally more efficient. Thus, developing efficient algorithms specializing in stochastic setting is important. Furthermore, as we will later describe, the stochastic setting we consider in this paper is not a special case of the adversarial settings that has been studied in the literature. Specifically, past research in the adversarial setting assume the reward function has extra properties that, when specialized to the stochastic setting, are overly restrictive.

When the reward depends non-linearly on the ground set, additional challenges have been added to develop efficient algorithms. For example, opening additional restaurants in a small market may result in diminishing returns due to market saturation. Such diminishing returns can be naturally modeled with the class of submodular set functions. A set function $f : 2^\Omega \to \mathbb{R}$ defined on a finite ground set $\Omega$ is said to be *submodular* if it satisfies the diminishing return property: for all $A \subseteq B \subseteq \Omega$, and $x \in \Omega \setminus B$, it holds that $f(A \cup \{x\}) - f(A) \geq f(B \cup \{x\}) - f(B)$ [Nemhauser et al.,

*Accepted for the 38th Conference on Uncertainty in Artificial Intelligence* (UAI 2022).

1978]. In this paper, we focus on the problem of combinatorial multi-armed bandits with stochastic submodular (in expectation) rewards and full-bandit feedback. We further assume that the reward function is monotone: a submodular set function $f : 2^{\Omega} \to \mathbb{R}$ is called monotone if for any $A \subseteq B \subseteq \Omega$ we have $f(A) \leq f(B)$.

## 1.1 MOTIVATING EXAMPLES

**Influence Maximization** Consider a case of social network where a company developed an application and wants to market it through the network. The best way to do this is selecting a set of highly influential users and hope they can love the application and recommend their friends to use it. Influence maximization is a problem of finding a small subset (seed set) in a network that can achieve maximum influence. This subset selection problem in social networks is commonly modeled as an offline submodular optimization problem [Domingos and Richardson, 2001, Kempe et al., 2003, Chen et al., 2010]. Algorithms and heuristics for solving this problem often assume knowledge of the network and diffusion model. A recent line of research has generalized the problem as a multi-armed bandit problem (with extra feedback) where the knowledge of the network and diffusion model is not required [Lei et al., 2015, Wen et al., 2017, Vaswani et al., 2017, Li et al., 2020, Perrault et al., 2020].

**Recommender Systems** When recommending bundles of items, such as movies, news articles, or consumer products, considering the estimated individual item rankings alone may be suboptimal. The system should recommend diversified items to maximize the coverage of information that users are interested, in order to get as much positive feedback as possible. This is motivated by recommending items with redundant information leads to diminishing returns on utility. This problem of sequentially recommending sets of items to users has been studied through the framework of contextual submodular combinatorial bandits [Qin and Zhu, 2013, Yue and Guestrin, 2011, Takemori et al., 2020].

**Crowdsourcing and Crowdsensing** Crowdsourcing involves batches of simple tasks being sequentially assigned to workers with unknown quality and speed. For example, workers may be recruited to manually label images in a database. Crowdsensing involves sequentially collecting data from large numbers of users in different locations. For instance, mobile phone accelerometer data can help identify potholes in city roads. Instances of these problems often involve sequential decision making of assigning/selecting subsets of workers/users with unknown qualities and under a budget. There is a line of research on this topic using the framework of combinatorial multi-armed bandits with submodular rewards [Zhang and van der Schaar, 2012, Nushi et al., 2016, Song and Jin, 2021].

## 1.2 OUR CONTRIBUTION

The main contribution of this paper can be summarized as follows:

- We propose Explore-then-Commit Greedy (ETCG), the first algorithm designed for stochastic CMAB problems with a submodular reward function (in expectation) and full-bandit feedback. It is procedurally simple and has low storage and per-round computational complexity.

- We prove that ETCG achieves $\mathcal{O}(n^{\frac{1}{3}} k^{\frac{4}{3}} T^{\frac{2}{3}} \log(T)^{\frac{1}{2}})$ expected cumulative $(1 - 1/e)$-regret.

- We show ETCG outperforms other full-bandit methods on experiments with synthetic and real-world data.

## 1.3 RELATED WORK

We now briefly discuss related works from several research topics that overlap in multiple aspects with the problem we study. Table 1 lists related works and enumerates aspects of the problem setup including properties of the reward function, the feedback model, and regret type. We let $n$ denote the number of base arms, $k$ the maximum cardinality, and $T$ the time horizon.

**Adversarial** The closest related works are those for adversarial CMAB with submodular rewards, full-bandit feedback, and cumulative regret. In the adversarial setting, the environment chooses a sequence of monotone and submodular functions $\{f_1, \ldots, f_T\}$. This is incompatible with our setting, since we only require the set function $f_t$ to be monotone and submodular *in expectation*. Regret in the adversarial setting is also different—the decision-maker competes against a maximizing action over the sum of the sequence, $(1 - 1/e) \max_{a \in \mathcal{A}} \sum_{t=1}^{T} f_t(a)$.

We nonetheless consider the following regret bounds to be relevant benchmarks for the stochastic setting.

Streeter and Golovin [2008] proposed an algorithm that achieves $\mathcal{O}(k^2 (n \log n)^{1/3} T^{2/3} (\log T)^2)$ $(1 - 1/e)$-regret. The method we will propose, ETCG, will have a lower regret bound, by a factor of $k^{2/3}$ (ignoring log terms). Golovin et al. [2014] later proposed an algorithm that achieves $\mathcal{O}(k^{2/3} n^{2/3} (\log n)^{1/3} T^{2/3})$ $(1 - 1/e)$-regret. Recently, Niazadeh et al. [2021] proposed a new algorithm for the adversarial setting that achieves $\mathcal{O}(k n^{2/3} (\log n)^{1/3} T^{2/3})$ $(1 - 1/e)$-regret. The method we will propose, ETCG, will have a much lower regret bound than those two, by a factor of $n^{1/3}$ for both (ignoring log terms), for problems where there are many base arms relative to the cardinality constraint (i.e. $n \gg k$), such as social influence maximization.

**Semi-bandit** To our knowledge, all prior works on stochastic, combinatorial multi-armed bandits with submod-

| | Reward | | Feedback | Regret | |
|---|:---:|:---:|:---:|:---:|:---:|
| | Submodular | Stochastic | Full-Bandit | Cumulative | $(1 - 1/e)$ Bound |
| Streeter and Golovin [2008] | ✓ | | ✓ | ✓ | $\tilde{\mathcal{O}}( n^{\frac{1}{3}} k^2 T^{\frac{2}{3}} )$ |
| Golovin et al. [2014] | ✓ | | ✓ | ✓ | $\tilde{\mathcal{O}}( n^{\frac{2}{3}} k^{\frac{2}{3}} T^{\frac{2}{3}} )$ |
| Niazadeh et al. [2021] | ✓ | | ✓ | ✓ | $\tilde{\mathcal{O}}( n^{\frac{2}{3}} k \ T^{\frac{2}{3}} )$ |
| Agarwal et al. [2021b] | | ✓ | ✓ | ✓ | $\tilde{\mathcal{O}}( n^{\frac{1}{2}} k^{\frac{3}{2}} T^{\frac{1}{2}} )$ |
| Agarwal et al. [2021a] | | ✓ | ✓ | ✓ | $\tilde{\mathcal{O}}( n^{\frac{1}{3}} k^{\frac{1}{2}} T^{\frac{2}{3}} )$ |
| Chen et al. [2018] | ✓ | ✓ | | ✓ | $\tilde{\mathcal{O}}(T^{\frac{1}{2}})^{\dagger}$ |
| Du et al. [2021] | | ✓ | ✓ | | —— |
| ETCG (ours) | ✓ | ✓ | ✓ | ✓ | $\tilde{\mathcal{O}}( n^{\frac{1}{3}} k^{\frac{4}{3}} T^{\frac{2}{3}} )$ |

Table 1: Table of select related works, enumerating which problem and performance aspects are shared with our proposed ETCG. The notation $\tilde{\mathcal{O}}(\cdot)$ drops $\log$ terms. $^{\dagger}$[Chen et al., 2018] require additional smoothness properties of $f$ and the dependence on $k$ and $n$ is unknown.

ular rewards assume semi-bandit feedback. In this setting, the decision maker receives additional feedback. For example, in [Lin et al., 2015], the decision maker receives not only the reward of the chosen subset but also learns marginal gains of its elements. Several methods have been proposed that solve a continuous optimization problem as a surrogate for the submodular set function and require gradient estimates through extra feedback [Zhang et al., 2019, Chen et al., 2018, Zhu et al., 2021]. The "linear submodular bandit" problem involves maximizing a linear combination of known submodular functions, with marginal gains provided as extra feedback [Yue and Guestrin, 2011, Yu et al., 2016, Takemori et al., 2020]. Research on the application of online influence maximization use extra feedback about the nodes and/or edges in the diffusion tree [Lei et al., 2015, Wen et al., 2017, Vaswani et al., 2017, Li et al., 2020, Perrault et al., 2020]. Streeter and Golovin [2008] and Niazadeh et al. [2021] also proposed algorithms for the adversarial setting using semi-bandit feedback, improving their respective $(1 - 1/e)$-regret bounds to $\mathcal{O}(\sqrt{kT \log(n)})$ and $\mathcal{O}(k\sqrt{T \log(n)})$, respectively.

**Continuous Submodular** There is an active area of research in (continuous) optimization for functions exhibiting diminishing returns properties analogous to (discrete) optimization of submodular set functions. Several methods have been proposed in the bandit setting, varying in the environment (adversarial/stochastic) and feedback model [Chen et al., 2018, 2020, Zhang et al., 2019, Hassani et al., 2017, Mokhtari et al., 2020, Hassani et al., 2020, Zhang et al., 2020]. Extensions of these methods to problems with discrete actions have been proposed, but require additional assumptions, semi-bandit feedback, or expensive sampling routines to estimate gradients.

**Pure Exploration** Instead of evaluating algorithms in terms of *cumulative* regret, the decision maker may seek to only evaluate the regret of the action chosen at time $T$,

allowing for more aggressive exploration, or to select an action within a pre-set level of confidence as quickly as possible. Several works have investigated this "pure exploration" setting with semi-bandit feedback [Chen et al., 2016, Mokhtari et al., 2018, Merlis and Mannor, 2019, Jourdan et al., 2021] and recently for full-bandit feedback [Du et al., 2021] (for a special reward function).

**Non-submodular** There are prior works for combinatorial MAB with stochastic rewards and full-bandit feedback, but the classes of the reward functions considered do not include submodular functions. In particular, there are works for linear reward functions [Dani et al., 2008, Rejwan and Mansour, 2020] and Lipschitz reward functions [Agarwal et al., 2021a,b]. For those classes of reward functions considered by Rejwan and Mansour [2020], Agarwal et al. [2021a,b], the optimal action (best set of $k$ arms) is to use the $k$ *individually best* arms; that property does not hold for submodular rewards.

## 2 PROBLEM STATEMENT

In this section, we will formally present the problem we will study. We consider sequential decision-making problems with a fixed time horizon $T$, where at each time step $t$, the learner selects a subset (action) $S_t \subseteq \Omega$ with cardinality at most $k$. Let $\Omega$ be the ground set of base arms, and let $n = |\Omega|$ denote the number of arms. We will use the terminologies *subset* and *action* interchangeably throughout the paper. Let $\mathcal{S} = \{S | S \subseteq \Omega \text{ and } |S| \leq k\}$ denote the set of all allowed subsets at any time step. After the subset $S_t$ is selected, the learner receives reward $f_t(S_t)$. We assume the reward $f_t$ is stochastic, bounded in $[0, 1]$, and i.i.d. conditioned on a given subset. Define the expected reward function as $f(S) = \mathbb{E}[f_t(S)]$. We assume $f(S)$ to be submodular and monotonically non-decreasing. The goal of the learner is to maximize the cumulative reward $\sum_{t=1}^{T} f_t(S_t)$.

To measure the performance of the algorithm, one common metric is to compare the learner to an agent with access to a value oracle for $f$. Let $S^* = \arg\max_{S:|S|\leq k} f(S)$ denote the optimal solution. Maximizing a monotone submodular set function under a cardinality constraint is NP-hard even with a value oracle. The best achievable approximation ratio with a polynomial time algorithm is $1 - 1/e$ [Nemhauser et al., 1978]. Thus, we compare the learner's cumulative reward to $(1 - 1/e)Tf(S^*)$ and we denote the difference as the $(1 - 1/e)$-regret $\mathcal{R}_{1-1/e,T}$:

$$\mathcal{R}_{1-1/e,T} := (1 - \frac{1}{e})Tf(S^*) - \sum_{t=1}^{T} f_t(S_t). \quad (1)$$

Note that the $(1 - 1/e)$-regret $\mathcal{R}_{1-1/e,T}$ is random, depending on the rewards and subsets chosen. In designing an algorithm, we will focus on minimizing the expected cumulative $(1 - 1/e)$-regret

$$\mathbb{E}[\mathcal{R}_{1-1/e,T}] = (1 - \frac{1}{e})Tf(S^*) - \mathbb{E}\left[\sum_{t=1}^{T} f_t(S_t)\right], \quad (2)$$

where the expectation is over both the environment the sequence of actions. For ease of notation, we write $\mathcal{R}_T$ for $\mathcal{R}_{1-1/e,T}$ throughout this paper.

**Remark 2.1.** For the experiments in Section 5, we will not know $S^*$ and so will not be able to compute the $(1 - 1/e)$ regret (2). We will instead compute an upper bound. We will compare ETCG and baselines against $T$ times the expected value $f(S^{\mathrm{grd}})$ of the solution $S^{\mathrm{grd}}$ returned from an offline (greedy) approximation algorithm [Nemhauser et al., 1978]. Since $f(S^{\mathrm{grd}}) \geq (1 - \frac{1}{e})f(S^*)$, the expected cumulative regret with respect to $S^{\mathrm{grd}}$ upper-bounds (2). When the inequality is strict, $f(S^{\mathrm{grd}}) > (1 - \frac{1}{e})f(S^*)$, it is possible that the expected cumulative regret (2) is sub-linear in the horizon $T$ while the expected cumulative regret with respect to $S^{\mathrm{grd}}$ is linear in the horizon $T$.

## 3 ETCG ALGORITHM

In this section, we present our proposed algorithm, *Explore-Then-Commit Greedy* (ETCG). The pseudo code for ETCG is presented in Algorithm 1. Our algorithm adds base arms to a super arm (subset of base arms) over time greedily until the cardinality constraint is satisfied and then exploits that super arm. Let $S^{(i)}$ denote the super arm when we have selected $i < k$ base arms. Our procedure begins with the empty set, $S^{(0)} = \emptyset$. After fixing a subset $S^{(i-1)}$ with $i - 1$ arms, our procedure explores base arms to add to $S^{(i-1)}$ for an interval of time we refer to as *phase $i$*. Our procedure repeats this process until the cardinality constraint $k$ is satisfied.

Let $T_i$ denote the time step when phase $i$ finishes, for $i \in \{1, \cdots, k\}$. For notational consistency, we also denote $T_0 =$

---

**Algorithm 1** Explore-then-Commit Greedy (ETCG)

**Input:** set of base arms $\Omega$, horizon $T$, cardinality constraint $k$
Initialize $S^{(0)} \leftarrow \emptyset, n \leftarrow |\Omega|$
Initialize $m \leftarrow \left\lceil \left(\frac{T\sqrt{2\log(T)}}{n+2nk\sqrt{2\log(T)}}\right)^{2/3} \right\rceil$
**for** *phase $i \in \{1, \ldots, k\}$* **do**
    **for** *arm $a \in \Omega \setminus S^{(i-1)}$* **do**
        Play $S^{(i-1)} \cup \{a\}$ $m$ times
        Calculate the empirical mean $\bar{f}(S^{(i-1)} \cup \{a\})$
    **end for**
    $a_i \leftarrow \arg\max_{a \in \Omega \setminus S^{(i-1)}} \bar{f}(S^{(i-1)} \cup \{a\})$
    $S^{(i)} \leftarrow S^{(i-1)} \cup \{a_i\}$
**end for**
**for** *remaining time* **do**
    Play action $S^{(k)}$
**end for**

---

0 and $T_{k+1} = T$. Let $\bar{f}_t(S)$ denote the empirical mean reward of set $S$ up to and including time $t$. Let

$$\mathcal{S}_i := \{ S^{(i-1)} \cup \{a\} : a \in \Omega \setminus S^{(i-1)} \}$$

denote the set of actions considered during phase $i$. Each action consists of the super arm $S^{(i-1)}$ decided during the last phase and an additional base arm. Each action $S \in \mathcal{S}_i$ will be played the same number of times; let $m$ denote that number. The choice of $m$ will be optimized later to minimize regret. At the end of phase $i \in \{1, \ldots, k\}$, ETCG will select the action that has the largest empirical mean,

$$a_i = \arg\max_{a \in \Omega \setminus S^{(i-1)}} \bar{f}_{T_i}(S^{(i-1)} \cup \{a\}), \quad (3)$$

and include it in the super arm $S^{(i)} = S^{(i-1)} \cup \{a_i\}$. During the final phase, the algorithm exploits $S^{(k)}$; it plays the same action $S_t = S^{(k)}$ for $t \in \{T_k + 1, \cdots, T\}$.

We note that for the special setting of deterministic rewards, the choice (3) corresponds to the classic offline greedy approximation algorithm proposed by Nemhauser et al. [1978]. When the rewards are stochastic, the actions selected by ETCG may differ from those that the greedy algorithm [Nemhauser et al., 1978] would choose using a value oracle for the set function $f$ of expected rewards.

ETCG has low storage complexity and per-round time-complexity. During exploitation, for $t \in \{T_k + 1, \ldots, T_{k+1}\}$, ETCG only needs to store the indices of the $k$ base arms and does not need any computation. During exploration, for $t \in \{1, \ldots, T_k\}$, ETCG just needs to update the empirical mean for the current action at time $t$ and store the highest empirical mean so far in the current phase $i$ and its associated base arm $a \in \Omega \setminus S^{(i)}$. Thus, ETCG has $\mathcal{O}(k)$ storage complexity and $\mathcal{O}(1)$ per-round time complexity. For comparison, the algorithm proposed by Streeter and

Golovin [2008] for the adversarial full-bandit setting uses $\mathcal{O}(nk)$ storage complexity and and $\mathcal{O}(n)$ per-round time complexity.

**Remark 3.1.** When the time horizon is not known, we can use geometric doubling trick to extend our result to an any-time algorithm. Essentially, we pick a geometric sequence $T_i = T_0 2^i$ for $i \in \{1, 2, \cdots\}$, where $T_0$ is a large enough number to let the algorithm initialize, and run our algorithm within time interval $T_{i+1} - T_i$ with a full restart. We refer to the general detailed procedure in Besson and Kaufmann [2018]. From Theorem 4 in Besson and Kaufmann [2018], we can show that the regret bound conserves the original $T^{2/3} \log(T)^{1/2}$ dependence with only changes in constant factors.

# 4 REGRET ANALYSIS

In this section, we analyze the regret for Algorithm 1. We begin by stating the main theorem, which bounds the cumulative expected $(1 - 1/e)$-regret:

**Theorem 4.1.** *For the sequential decision making problem defined in Section 2 with $T \geq n(k + 1)$, the expected cumulative $(1 - 1/e)$-regret of ETCG is at most $\mathcal{O}(n^{\frac{1}{3}} k^{\frac{4}{3}} T^{\frac{2}{3}} \log(T)^{\frac{1}{2}})$.*

The detailed proof is in the supplementary material. We next briefly walk through the proof, highlighting some unique steps.

Since for each phase $i$, we play each action $S^{(i-1)} \cup \{a\} \in \mathcal{S}_i$ exactly $m$ times, we consider the equal-sized confidence radii $\mathrm{rad} := \sqrt{2\log(T)/m}$ for all the actions $S^{(i-1)} \cup \{a\} \in \mathcal{S}_i$ at the end of phase $i$. Denote the event that the empirical means of actions played in phase $i$ are concentrated around their statistical means as

$$\mathcal{E}_i := \bigcap_{S \cup \{a\} \in \mathcal{S}_i} \left\{ |\bar{f}(S \cup \{a\}) - f(S \cup \{a\})| < \mathrm{rad} \right\}. \quad (4)$$

Then we define the *clean event* $\mathcal{E}$ to be the event that the empirical means of all actions played up to and including phase $k$ are within $\mathrm{rad}$ of their corresponding statistical means:

$$\mathcal{E} := \mathcal{E}_1 \cap \cdots \cap \mathcal{E}_k. \quad (5)$$

Although the $\mathcal{E}_i$'s are not independent, by conditioning on the sequence of selected subsets $\{S^{(0)}, S^{(1)}, \ldots, S^{(k)}\}$ and using the Hoeffding bound, we show $\mathcal{E}$ happens with high probability. We then use the concentration of empirical means (4) and properties of submodular set functions to show the following important lemma.

**Lemma 4.2.** *Under the clean event $\mathcal{E}$, for all $i \in \{1, 2, \cdots, k\}$,*

$$f(S^{(i)}) - f(S^{(i-1)}) \geq \frac{1}{k}\left[f(S^*) - f(S^{(i-1)})\right] - 2\mathrm{rad}.$$

This lemma (Lemma 1.3 in the supplementary material) identifies a lower bound of the expected marginal gain $f(S^{(i)}) - f(S^{(i-1)})$ of the empirically best action $S^{(i)}$ at the end of phase $i$. The sequence of subsets $\{S^{(0)}, S^{(1)}, \ldots, S^{(k)}\}$ that ETCG picks *does not necessarily match* the sequence chosen by the offline greedy approximation [Nemhauser et al., 1978] using a value oracle for the expected reward function $f$. Even though ETCG may select a different sequence, Lemma 4.2 ensures the expected marginal gain is not too small. As a corollary of Lemma 4.2, using properties of submodular set functions and unraveling the recursion induced by Lemma 4.2, we can lower bound the expected value of ETCG's chosen set $S^{(k)}$ of size $k$, which is used for exploitation in phase $k + 1$:

**Corollary 4.3.** *Under the clean event $\mathcal{E}$,*

$$f(S^{(k)}) \geq (1 - \frac{1}{e})f(S^*) - 2k\mathrm{rad}. \quad (6)$$

This corollary appears as Corollary 1.4 in the supplementary material in Section 1.1.

Using Corollary 4.3, we can break up the expected $(1 - \frac{1}{e})$-regret (2) conditioned on the clean event $\mathcal{E}$ into two parts, one part for the first $k$ phases and one part for the exploitation phase,

$$\mathbb{E}[\mathcal{R}(T)|\mathcal{E}]$$
$$= (1 - \frac{1}{e})Tf(S^*) - \sum_{t=1}^{T} \mathbb{E}[f_t(S_t)]$$
$$= \sum_{t=1}^{T}\left((1 - \frac{1}{e})f(S^*) - \mathbb{E}[f(S_t)]\right)$$
$$= \underbrace{\sum_{i=1}^{k} \sum_{t=T_{i-1}+1}^{T_i}\left((1 - \frac{1}{e})f(S^*) - \mathbb{E}[f(S_t)]\right)}_{\text{First } k \text{ phases (exploration)}}$$
$$+ \underbrace{\sum_{t=T_k+1}^{T}\left((1 - \frac{1}{e})f(S^*) - \mathbb{E}[f(S^{(k)})]\right)}_{\text{Phase } k + 1 \text{ (exploitation)}}. \quad (7)$$

Recall that in phase $i$, each of the $n - (i - 1)$ actions in $\mathcal{S}_i$ is played exactly $m$ times, meaning $T_i - T_{i-1} = m(n-i+1)$. For each action $S_t$ played during phase $i$, that is for $t \in \{T_{i-1}+1, \cdots, T_i\}$, since $S^{(i-1)} \subset S_t$, by monotonicity of the expected reward function $f$ we have $f(S^{(i-1)}) \leq f(S_t)$. Thus we can upper bound the expected regret $\mathbb{E}[\mathcal{R}(T)|\mathcal{E}]$

incurred during the first $k$ phases (first term of (7)) as

$$\sum_{i=1}^{k} \sum_{t=T_{i-1}+1}^{T_i} \left( (1 - \frac{1}{e})f(S^*) - \mathbb{E}[f(S_t)] \right)$$

$$\leq \sum_{i=1}^{k} m(n-i+1) \left( (1 - \frac{1}{e})f(S^*) - \mathbb{E}[f(S^{(i-1)})] \right)$$

$$\leq mn \sum_{i=1}^{k} \left( (1 - \frac{1}{e})f(S^*) - \mathbb{E}[f(S^{(i-1)})] \right). \qquad (8)$$

We can further upper bound (8) as

$$\sum_{i=1}^{k} \left( (1 - \frac{1}{e})f(S^*) - \mathbb{E}[f(S^{(i-1)})] \right)$$

$$\leq \sum_{i=1}^{k} \left( f(S^*) - \mathbb{E}[f(S^{(i)})] \right)$$

$$\leq k \sum_{i=1}^{k} \left( \mathbb{E}[f(S^{(i)})] - \mathbb{E}[f(S^{(i-1)})] + 2\mathrm{rad} \right) \qquad (9)$$

$$= k(\mathbb{E}[f(S^{(k)})] - \mathbb{E}[f(S^{(0)})] + 2k\mathrm{rad}) \qquad (10)$$

$$\leq k \left( 1 + 2k\mathrm{rad} \right), \qquad (11)$$

where (9) follows by applying Lemma 4.2 and taking expectation, (10) follows by simplifying a telescoping sum, and (11) by $\mathbb{E}[f(S^{(k)})] \leq 1$ and $\mathbb{E}[f(S^{(0)})] = 0$.

We can upper bound the expected regret $\mathbb{E}[\mathcal{R}(T)|\mathcal{E}]$ incurred during the exploitation phase (phase $k+1$; second term of (7)) by applying Corollary 4.3 as

$$\sum_{t=T_k+1}^{T} \left( (1 - \frac{1}{e})f(S^*) - \mathbb{E}[f(S^{(k)})] \right)$$

$$\leq \sum_{t=T_k+1}^{T} 2k\mathrm{rad} \leq 2kT\mathrm{rad}. \qquad (12)$$

Combining the upper bounds (11) and (12) and then optimizing over the number of times $m$ each action is sampled during exploration, we get

$$\mathbb{E}[\mathcal{R}(T)|\mathcal{E}]$$

$$\leq 4n^{\frac{1}{3}} k (T\sqrt{2\log(T)})^{\frac{2}{3}} (1 + 2k\sqrt{2\log(T)})^{\frac{1}{3}}$$

$$= \mathcal{O}(n^{\frac{1}{3}} k^{\frac{4}{3}} T^{\frac{2}{3}} \log(T)^{\frac{1}{2}}). \qquad (13)$$

We then show that because the clean event $\mathcal{E}$ happens with high probability, $\mathbb{E}[\mathcal{R}(T)]$ also satisfies (13), completing the proof.

**Lower bounds:** For the setting we explore in this paper, with stochastic CMAB with submodular expected rewards

and full-bandit feedback, it remains an open question if $\tilde{\mathcal{O}}(T^{1/2})$ expected cumulative $(1 - 1/e)$-regret is possible (ignoring $n$ and $k$ dependence). For the special sub-class of linear reward functions, $\tilde{\Omega}(T^{1/2})$ is known [Dani et al., 2008].

# 5 EXPERIMENTS

We next evaluate our proposed algorithm ETCG on both synthetic data and real world data.

For the experiments, instead of $(1 - 1/e)$ regret Equation (1), which requires knowing $S^*$, we compare the cumulative rewards achieved by ETCG and baselines against $Tf(S^{\mathrm{grd}})$, where $S^{\mathrm{grd}}$ is the solution returned by the offline $(1-1/e)$-approximation algorithm proposed by Nemhauser et al. [1978]. Recall from Remark 2.1 that $Tf(S^{\mathrm{grd}}) \geq (1 - 1/e)Tf(S^*)$, so $Tf(S^{\mathrm{grd}})$ is a more challenging reference value.

## 5.1 BASELINE METHODS

We use three algorithms designed for CMAB with full-bandit feedback as baselines.

- **Online Greedy with opaque feedback model (OG$^o$)** [Streeter and Golovin, 2008] This algorithm is designed for the adversarial setting with submodular rewards. The adversary model is *oblivious*, meaning the sequence of monotone submodular reward functions is fixed in advance. OG$^o$ utilizes $k$ subroutines of randomized weighted majority algorithms [Littlestone and Warmuth, 1994] to select actions, where $k$ is the cardinality constraint. At each time step, the algorithm explores with probability $\gamma$ and exploits with probability $1 - \gamma$. During exploration, it randomly picks an randomized weighted majority subroutine to select a base arm to explore. OG$^o$ has an $\widetilde{\mathcal{O}}(T^{2/3})$ theoretical guarantee for the adversarial setting. We refer to our detailed implementation and parameter selection in Section 2.

- **CMAB-SM** [Agarwal et al., 2021a] This algorithm assumes the expected reward functions are Lipschitz continuous functions of individual arm rewards. The algorithm divides all $n$ base arms in to groups, sorts arms within each group, and then merges groups one by one to obtain the best $k$ arms. CMAB-SM has an $\widetilde{\mathcal{O}}(T^{2/3})$ theoretical guarantee.

- **DART** [Agarwal et al., 2021b] DART is a successive accept-reject style algorithm designed for Lipschitz reward functions that have an additional property related to the marginal gains of the base arms. DART has an $\widetilde{\mathcal{O}}(T^{1/2})$ theoretical guarantee.

## 5.2 EXPERIMENTS WITH SYNTHETIC DATA

We begin with experiments with two special cases of submodular set functions. The first one is mean (linear) functions of individual arm rewards $f_t(S) = \sum_{a \in S} f_t(\{a\})/k$. The second is a stochastic weighted set cover, which can be viewed as a simple model for product recommendations. Let $n$ denote the number of products and each product belongs to exactly one of $c$ different categories. These product categories also have different (expected) values given by the weight vector $\omega$. The expected instantaneous reward is defined as the average (over cardinality $k$) weight of the categories covered by a chosen set of up to $k$ products. With $C_i$ denoting indices of arms belonging to category $i$ and $\omega_t[i]$ denoting the instantaneous weight of category $i$ at time $t$, and $\mathbf{1}.$ denoting the indicator function, $f_t(S) = \frac{1}{k} \sum_{i=1}^{c} w_t[i] \mathbf{1}_{S \cap C_i \neq \emptyset}$. This reward function is monotone and submodular. Notice that for these two types of reward functions, the offline greedy solution is the optimal solution so we are actually comparing against the optimal solution in the results.

### 5.2.1 Experiment Details

For both setups, we use $n = 20$ base arms. The cardinality constraint is $k = 4$. We run experiments on different time horizons $T \in \{10^2, 10^3, 10^4, 10^5, 10^6\}$. For each horizon $T$ and reward function type (linear or weighted cover), we run each method 10 times.

For the linear reward function, for each run we first generate expected rewards $\{f(\{a\})\}_{a \in \Omega}$ for individual arms randomly $f(\{a\}) \overset{i.i.d.}{\sim} \mathcal{U}([0.1, 0.9])$. For each arm $a \in \Omega$, the instantaneous reward $f_t(\{a\})$ at time $t$ is the expected reward plus noise, $f_t(\{a\}) = f(\{a\}) + \epsilon_{a,t}$, where the noises $\{\epsilon_{a,t}\}_{a \in \Omega, 1 \leq t \leq T}$ are i.i.d. and follow a truncated normal distribution with mean 0 and standard deviation 0.1 within interval $[-0.1, 0.1]$ (so all instantaneous rewards $f_t(\cdot)$ are within the interval $[0, 1]$).

For the weighted cover problem, we used $c = 4$ categories with $[6, 6, 6, 2]$ products respectively. The stochastic weights for each category $i = 1, 2, 3, 4$ at time $t$ are drawn from a uniform distribution $\omega[i] \sim \mathcal{U}([0, i/5])$.

### 5.2.2 Results and Discussion

Figures 1a and 1b depict cumulative regret curves for ETCG (in blue) and baselines for different horizon $T$ values for the linear and weighted cover problems respectively. The standard deviation is also represented by error bars in the plots, though some of them might be hard to notice since the values of them are small. Figures 1c and 1d depict instantaneous rewards over a horizon $T = 10^5$ for linear and max rewards respectively. The curves are averaged over the

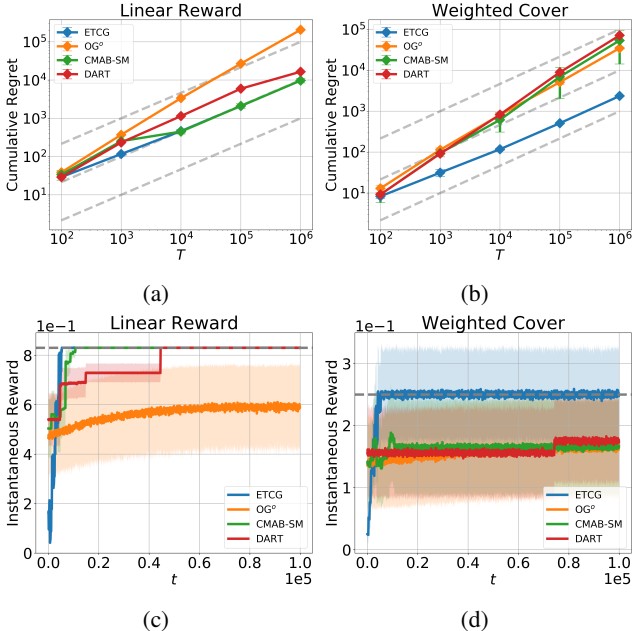

Figure 1: (a) and (b) are comparison results for cumulative regret as a function of time horizon $T$. (c) and (d) are the moving average plot with window size 100 of instantaneous reward as a function of $t$. The expected reward used in (a) and (c) is linear, and weighted cover reward is used in (b) and (d). The gray dashed lines in (a) and (b) represent $y = aT^{2/3}$ for various values of $a$. The gray dashed line in (c) and (d) represents the value of the optimal solution (averaged across runs).

10 runs. The shaded area is the standard deviation for each method. The instantaneous reward curves for all methods are smoothed with a moving average with window size 100. The gray dashed lines in Figures 1a and 1b represent $y = aT^{2/3}$ for various values of $a$, corresponding to cumulative regret curves of $\widetilde{O}(T^{2/3})$.

**Results–Linear** Recall that ETCG, OG$^\circ$, and CMAB-SM all have $\widetilde{O}(T^{2/3})$ regret (for their respective settings, which include linear functions). DART has $\widetilde{O}(T^{1/2})$ regret for this setting.

In Figure 1a, we can see ETCG (in blue) outperforms OG$^\circ$ (in orange) and DART (in red), and shares similar performance with CMAB-SM (in green). Over the horizons examined (up to $T = 10^6$), OG$^\circ$'s cumulative regret appears to grow faster than $T^{2/3}$ (i.e. the curve's slope appears steeper than $2/3$ on a log-log plot). One of the major reasons for this is that OG$^\circ$ explores actions (including actions will cardinality smaller than $k$) with a constant probability. Figure 1c shows that behavior also results in larger standard deviation area in the instantaneous reward curve and slower improvement in its instantaneous rewards.

**Results–Weighted Cover** Figure 1b shows the cumulative regret curve for the weighted cover problem. ETCG (in blue) outperforms all baseline methods by a large margin for all time horizons. Similar to what we have mentioned in linear case, we believe that OG$^o$ (in orange) performs poorly in part due to time spent in exploration.

DART's cumulative regret (in red) empirically grows as $O(T^{0.90})$, much faster than ETCG's growth of $O(T^{0.58}) < O(T^{2/3})$ (we empirically estimated the slopes of the regret curves for these methods on the log-log scale). CMAB-SM's cumulative regret curve (in green) grows almost as fast as DART's, indicating CMAB-SM and DART fail to select a good action. They work well in the linear case mainly because the assumptions for ETCG, CMAB-SM and DART are all satisfied, so the regret bound would hold. However, in weighted cover problem, unlike linear function, the reward function is not simply a function of individual base arm rewards, a property used by DART and CMAB-SM. The reward function exhibits arm set dependence.

## 5.3 EXPERIMENTS WITH REAL WORLD DATA

We next run experiments for the application of social network influence maximization over a portion of the Facebook network graph. While there are prior works proposing algorithms for influence maximization bandit problems, the state of the art (e.g., [Wen et al., 2017]) presumes knowledge of the diffusion model (such as independent cascade) and, more importantly, extensive semi-bandit feedback on individual diffusions, such as which specific nodes became active or along which edges successful infections occurred, in order to estimate diffusion parameters. For social networks with user privacy, this information is not available.

### 5.3.1 Data Set Description and Experiment Details

We next conduct experiments on an influence maximization problem using a portion of the Facebook network [Leskovec and Mcauley, 2012]. To facilitate running multiple experiments for different horizons, we used the community detection method proposed by Blondel et al. [2008] to detect a community with 534 nodes and 8158 edges. The diffusion process is simulated using the independent cascade model [Kempe et al., 2003], where in each discrete step, an active node (that was inactive at the previous time step) independently attempts to infect each of its inactive neighbors. We used uniform infection probabilities (0.1 for each edge). For each horizon $T \in \{2 * 10^4, 4 * 10^4, \dots, 10^5\}$, we tested each method ten times.

### 5.3.2 Results and Discussion

Figures 2a to 2c show average cumulative regret curves for ETCG (in blue) and baselines for different horizon $T$ values

when the cardinality constraint $k$ is 4, 8 and 16, respectively. The shaded areas depict the standard deviation. The figure axes are linearly scaled, so a linear cumulative regret curve corresponds to (linear) $\widetilde{O}(T)$ cumulative regret.

ETCG significantly outperforms OG$^o$ (in orange). Over the horizons tested, OG$^o$'s cumulative regret (averaged over ten runs) appears to grow linearly with $T$. We saw in Section 5.2 that even for much simpler reward functions and with few arms $n$ and small cardinality $k$, OG$^o$ performed poorly.

ETCG outperforms CMAB-SM (in green) for all time horizons and cardinalities, with significant gaps between ETCG and CMAB-SM for smaller $k$. From Figures 2a to 2c, CMAB-SM's performance appears fairly stable across increasing cardinalities (though note limits of y-axes differ) while ETCG's regret curve appears to grow (relative to others). For a fixed horizon $T$, increasing $k$ means more phases, which (for this problem with large $n$) means more time exploring overall but less time in any one phase, so the arms selected may not be as good. This phenomenon is visually apparent in the instantaneous reward plots Figures 2d to 2f. In Figure 2d with $k = 4$, for instance, each of the four phases of ETCG's exploration are visually distinct, and exploitation begins around $t = 20000$. In Figure 2f with $k = 16$, however, each of the sixteen phases of ETCG's exploration are shorter and exploitation begins around $t = 35000$.

ETCG and DART (in red) have similar performance for small time horizons. However, DART's cumulative regret curve has a steep jump which make the performance significantly worse. We attribute these jumps to the exponential epochs lengths considered in DART with number of epochs $\lfloor \log_2(KT/N \log(NT)) \rfloor$. This creates a non-smooth behavior in the regret growth of the DART algorithm.

Figure 2d, Figure 2e and Figure 2f shows instantaneous rewards over a horizon $T = 10^5$ for corresponding cardinality constraints. Again curves for all methods are smoothed with a moving average with window size 100. Clearly we can see that ETCG has the fastest convergence over all methods. On the other hand, the set of size $k$ that is chosen by ETCG is worse that those of CMAB-SM and DART, since the latter two methods requires longer time to explore. We can also attribute the worse performance when $k$ gets larger to the larger $k$ term in the regret bound.

## 6 CONCLUSION

In this paper, we investigate the problem of combinatorial multi-armed bandits in stochastic setting with expected rewards being submodular, where the agent can choose up to $k$ out of $n$ arms in each time step and receives only the aggregated reward. We proposed a simple algorithm ETCG, and showed that the algorithm is efficient both theoretically and empirically. We showed that it can achieve $\tilde{\mathcal{O}}(T^{\frac{2}{3}})$ $(1 - 1/e)$-regret, which is the first theoretical regret bound

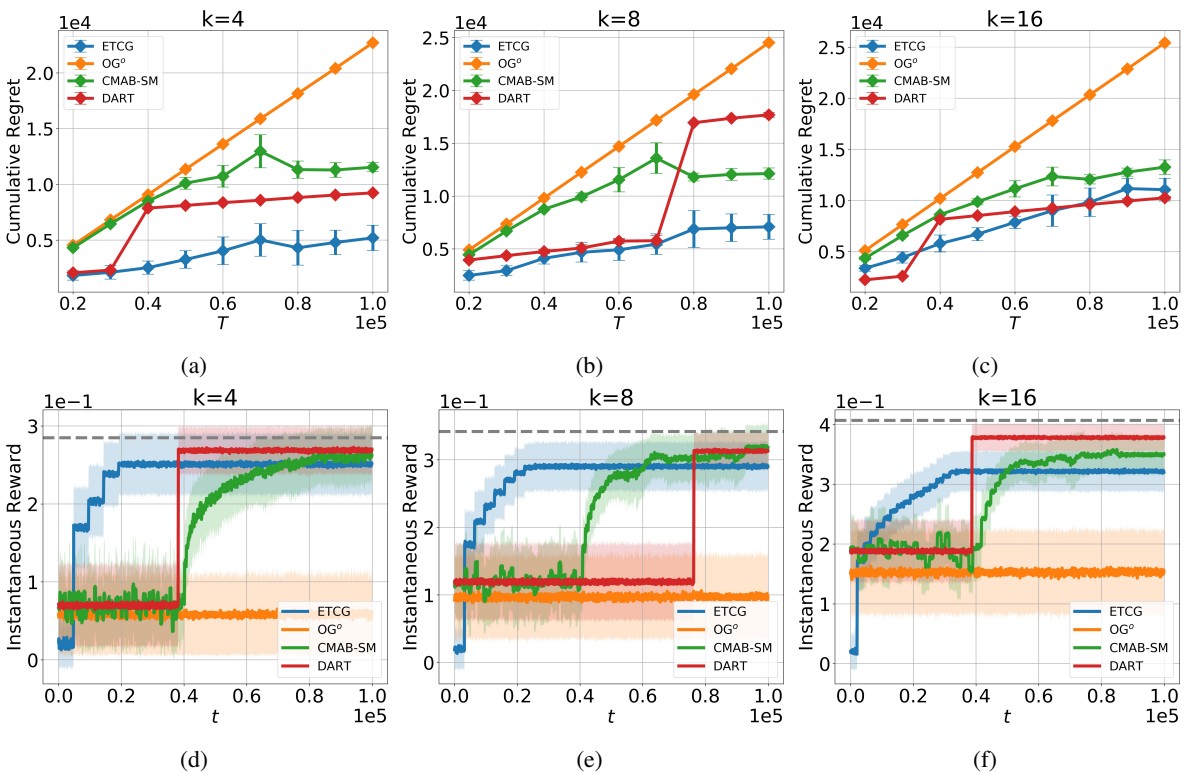

Figure 2: (a), (b) and (c) are comparison results for cumulative regret as a function of time horizon $T$. (d), (e) and (f) are the moving average plot with window size 100 of instantaneous reward as a function of $t$. The gray dashed lines in (d), (e) and (f) represent expected rewards for the action chosen by an offline greedy algorithm.

in stochastic, full-bandit, submodular reward settings, and is comparable to guarantees in adversarial settings evaluated in Streeter and Golovin [2008] and Niazadeh et al. [2021]. We empirically showed that it outperforms other baselines on synthetic data and on a social influence maximization network.

## Acknowledgements

This material is based upon work supported by the National Science Foundation under Grants No. 2149588 and 2149617.

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
