# OpenReview forum: "An Explore-then-Commit Algorithm for Submodular Maximization Under Full-bandit Feedback"
_auai.org/UAI/2022/Conference — UAI 2022 Poster_

### Official Review · Reviewer_BFxJ · 2022-04-01

**Q2(1) Originality/Novelty:** 2
**Q2(2) Significance/Impact:** 3
**Q2(3) Correctness/Technical Quality:** 3
**Q2(6) Clarity Of Writing:** 3
**Q6 Overall Score:** 5
**Q8 Confidence In Your Score:** 3

**Q1 Summary And Contributions:**

This paper focused on the combinatorial multiarmed bandit problem (CMAB) with submodular reward and full-bandit feedback. Compared with previous works, this paper aims at the stochastic reward function and proposes the ETCG algorithm with $O(n^{1/3}k^{4/2}T^{2/3})$ regret guarantee. In addition, the experiment results also support the efficiency of this novel algorithm.

**Q2 Assessment Of The Paper:**

More detailed information regarding each of these aspects is given below:

**Q2(4) Quality Of Experiments (Optional):**

3: Good: The experimental evaluation is adequate, and the results convincingly support the main claims.

**Q2(5) Reproducibility:**

3: Good: Key resources (e.g., proofs, code, data) are available and key details (e.g., proofs, experimental setup) are sufficiently well-described for competent researchers to confidently reproduce the main results.

**Q3 Main Strengths:**

1. The space complexity of the ETCG algorithm is only $O(k)$, and the time complexity within each round is only $O(1)$. Compared with the previous algorithm with space complexity $O(nk)$ and time complexity $O(n)$ within each round, the ETCG is much more efficient.

2. Theoretical result shows that the ETCG algorithm obtains a $O(n^{1/3}k^{4/2}T^{2/3})$ regret guarantee. Compared with previous results, this improves the dependency of factor $n$ or $k$.

3. Experiment results also show that the ETCG algorithm outperforms other baseline algorithms.

**Q4 Main Weakness:**

1. This paper only focused on the CMAB problem with the stochastic reward function, which seems much easier than the previous adversarial reward function setting. Though the author mentions that the stochastic reward function cannot be fully covered by the adversarial reward, it still seems that previous results can be directly used in the stochastic reward function with slight changes. Therefore, it seems not fair to directly compare with previous results on adversarial reward function setting.

2. The previous offline local greedy algorithm (Nemhauser et al. [1978]) can find a $(1-1/e)$-approximate near-optimal policy with the known reward function. The ETCG algorithm just combines the offline local greedy algorithm with a widely used exploration strategy within the multi-arm bandit problem to deal with the stochastic reward. Therefore, this paper seems an incremental work, and the novelty is limited.

**Q5 Detailed Comments To The Authors:**

1. Lemma 4.2 seems non-necessary. Since the function $f$ is bounded in the range $[0,1]$, the optimal function always satisfies that $f(S^*)\leq 1$. Therefore, the regret in equation (11) can be directly controlled by $k\times (1-1/e)$ without the help of Lemma 4.2.

2. For the experiments in section "EXPERIMENTS WITH SYNTHETIC DATA" with size $n=10,k=2$, it seems easy to compute the optimal policy by enumeration. Therefore, it looks better if the author can compare the optimal policy directly.

3. For Table 1, since the adversarial reward seems much more complicated than the stochastic reward, adding a new column with the attribute "adversarial reward" is more reasonable. In addition, though the stochastic reward function cannot be fully covered by the adversarial reward, it still seems that previous results can be directly used in the stochastic reward function with slight changes. Therefore, it is better if the author can talk more about whether the previous result in adversarial reward will fail in these cases.

**Q7 Justification For Your Score:**

Compared with the previous analysis, the ETCG algorithm obtains a$O(n^{1/3}k^{4/2}T^{2/3})$ regret guarantee, which improves the dependency of factor $n$ or $k$. In my view, the main result is significant.
However, this work only focused on the stochastic reward function. Moreover, the ETCG algorithm combines the offline local greedy algorithm with a widely used exploration strategy within the multi-arm bandit problem, limiting the novelty.

**Q9 Complying With Reviewing Instructions:**

1: Yes.

---

### Official Review · Reviewer_a6RR · 2022-04-06

**Q2(1) Originality/Novelty:** 2
**Q2(2) Significance/Impact:** 2
**Q2(3) Correctness/Technical Quality:** 4
**Q2(6) Clarity Of Writing:** 4
**Q6 Overall Score:** 6
**Q8 Confidence In Your Score:** 4

**Q1 Summary And Contributions:**

The paper studies the regret minimization task in stochastic combinatorial bandits with full-bandit feedback, where the expected rewards are monotone and submodular. An explore-then-exploit algorithm is suggested, for which theoretical guarantees in the form of an upper bound on its expected cumulative regret are shown. In an experimental study on synthetic as well as real-world data, the suggested algorithm is compared with existing methods designed for similar problem settings.


**Q2 Assessment Of The Paper:**

More detailed information regarding each of these aspects is given below:

**Q2(4) Quality Of Experiments (Optional):**

2: Fair: The experimental evaluation is weak: important baselines are missing, or the results do not adequately support the main claims.

**Q2(5) Reproducibility:**

2: Fair: Key resources (e.g., proofs, code, data) are unavailable but key details (e.g., proof sketches, experimental setup) are sufficiently well-described for an expert to confidently reproduce the main results.

**Q3 Main Strengths:**

- Except for minor things, the paper is well written and has a reasonable structure. Also it provides a good overview of the relevant literature.

-  The suggested approach is simple, yet has compared to existing methods a more favorable dependency on some problem-relevant parameters in the regret bounds. The proofs are very clearly presented and seem to be flawless.

- The considered (combinatorial) bandit problem is interesting from a practical perspective, as submodularity of the target function is oftentimes a reasonable assumption.



**Q4 Main Weakness:**

- The algorithm needs knowledge of the timehorizon.

- The experiments on synthetic data are not convincing, as DART eventually gets better than the suggested algorithm if the time horizon is large enough, although it’s theoretical guarantees do not necessarily hold for the considered submodular rewards. It would have been better to consider a scenario with submodular expected rewards, where DART cannot catch up.

- As the problem setting has been considered by Du et al. 2021 in a pure exploration setting with fixed confidence, there is already an explore-then-exploit algorithm available for this setting: Explore with the algorithm(s) of Du et al. 2021 with confidence level $\delta = 1/T$ and exploit then the returned super arm. This already provides another theoretical guarantee in Table 1 and another baseline for the experiments.

- The paper leaves open the tempting question of a tight lower bound on the expected regret for any learning algorithm in this environment.


**Q5 Detailed Comments To The Authors:**

*Choice of the confidence radius*: I was wondering about the choice of the confidence radius $rad,$ as it would suffice to use a radius such that the probability of the bad event $\mathcal{E}^c$ is bounded by $Cnk/T$ for some constant $C>0.$ This could be achieved by using $rad = \sqrt{  \frac{\log(T)}{2m} }.$ But perhaps this would lead to difficulties for the analysis of the optimality of $m$ on p.17 and 18. Maybe the authors can comment on this.

*Experiments*: It would have been more interesting to investigate the sensitivity of the empirical regrets of the considered methods with respect to the number of base arms $n$ and the subset size $k.$ In this way one could see the whether the dependency on these terms are ''right’’ in the corresponding bound.

**Other minor things:**

- Introduction: There are some missing ''the’’ ‘s in this section, e.g. “extension of the MAB problem … to the selected super arm … satisfies the diminishing return property”. Also there is a typo: “designes”.


- Table 1: Perhaps you could add another column with ''further properites’’ or ''additional properties’’, where your approach has the *monotonicity* and the ones by Agrawal et al. *Lipschitz*.

- p.4 (left column): `` … it is possible [that] the … ‘’

- p.5 (before Eq.(4)): `` … concentrated [around] …’’

- p.6 - 7 (a before O-term): it should be ‘’an’’ before the O-terms, respectively

- p.7 (left column): ``… DART up to $T=10^6$.”

- Lemma A.2: The clean event $\mathcal{E}$ [in] (16) …

- In the appendix there is repeatedly the remark on the fact that ETCG does not necessarily picks the same sequence of subsets as the offline greedy approximation by Nemheuser et al. I think it is enough to state this before Corollary A.4.

- I am not sure whether the known properties in Section B need to be proved.

# Post Rebuttal

After reading the other reviews as well as the author's response, I am even more inclined to accept the paper, as I think the authors did a good and convincing job in their rebuttal. I appreciate the efforts of the authors for having conducted new and even more convincing experiments. Moreover, thanks to Reviewer BFxJ, the theoretical guarantee of their algorithm has become even tighter w.r.t. $k.$

Considering that the paper has already been carefully written, I am not worried that the authors will adequately incorporate the new experiments and the improved theoretical result.



**Q7 Justification For Your Score:**

The paper has pros and cons as described above. Although the experimental study as well as the mechanism of the algorithm leave room for improvements, I am slightly in favor of accepting the paper due to its clear presentation and the practical relevance of the problem. However, I think the algorithm can be improved in many ways (not knowing the timehorizon, exploration lengths of the phases).

**Q9 Complying With Reviewing Instructions:**

0: No.

---

### Official Review · Reviewer_H3aE · 2022-04-06

**Q2(1) Originality/Novelty:** 3
**Q2(2) Significance/Impact:** 3
**Q2(3) Correctness/Technical Quality:** 3
**Q2(6) Clarity Of Writing:** 4
**Q6 Overall Score:** 6
**Q8 Confidence In Your Score:** 2

**Q1 Summary And Contributions:**

The authors consider the combinatorial multi-armed bandit with stochastic submodular rewards, and propose a new algorithm, showing that it outperforms other methods in a number of experiments.

**Q2 Assessment Of The Paper:**

More detailed information regarding each of these aspects is given below:

**Q2(4) Quality Of Experiments (Optional):**

3: Good: The experimental evaluation is adequate, and the results convincingly support the main claims.

**Q2(5) Reproducibility:**

3: Good: Key resources (e.g., proofs, code, data) are available and key details (e.g., proofs, experimental setup) are sufficiently well-described for competent researchers to confidently reproduce the main results.

**Q3 Main Strengths:**

The paper is well-written and easy to follow. The authors make a fair discussion of other models in the literature and introduce the problem and its connections with the existing work. The idea behind the algorithm is fairly simple, but it seems to perform quite well.

**Q4 Main Weakness:**

Perhaps a minor weakness would be the lack of a discussion about the extendability of this approach to other scenarios. In this sense, I wonder if (i) the idea of selecting at each step the action with highest empirical term can be used with other algorithms and (ii) if other measures instead of the mean could also be useful.

From the results section in Figure 1, it seems that the DART algorithm performs better than the ETCG for (very) large values of T, but this fact is only mentioned but not discussed further. Something similar happens with the results discussed in Figure 2, although some comments are given in that case. It would perhaps be interesting to discuss whether the algorithm can be modified so as to join the relative strenghts of ETCG and DART.

**Q5 Detailed Comments To The Authors:**

In addition to the comments above, I have a few other minor suggestions:

1) Some properties of submodularity are used in the proofs in the supplementary material, but it would be interesting to know how essential they are, as it may allow to consider an alternative to the models mentioned at the end of section 1.

2) A section with conclusions and future lines of work should be added.

3) There are a few minor typos:

-Page 2, first column: a dot is missing between 'feedback' and 'We further'.
-Page 6, second column: you mention four algorithms but only discuss three. Also, 'it randomly pick' (should be 'picks')
-Page 7, first column: 'outperform' -> 'outperforms'.
-Page 8, second column: 'shows' -> 'show'.

**Q7 Justification For Your Score:**

The paper is well written, clearly explained, and the algorithm performs well. Still, it suffers in the comparison with DART.

**Q9 Complying With Reviewing Instructions:**

1: Yes.

---

### Official Review · Reviewer_L6b1 · 2022-04-16

**Q2(1) Originality/Novelty:** 2
**Q2(2) Significance/Impact:** 2
**Q2(3) Correctness/Technical Quality:** 2
**Q2(6) Clarity Of Writing:** 2
**Q6 Overall Score:** 5
**Q8 Confidence In Your Score:** 3

**Q1 Summary And Contributions:**

This paper looks at the combinatorial multi-arm bandit problem and proposes an algorithm based on maximization of submodular functions. In particular, the paper shows the (1-1/e)-regret upper bound of O(n^(⅓)k^(4/3)T^(⅔)log(T^(½)) for a horizon T. Experiments are shown on synthetic and real datasets and the comparison is done with several baselines such as DART [Agarwal et al. 2021b], CMAB-SM [Agarwal et al. 2021a] , submodular online greedy (OG) [ Streeter and Golovin, 2008].

**Q2 Assessment Of The Paper:**

More detailed information regarding each of these aspects is given below:

**Q2(4) Quality Of Experiments (Optional):**

2: Fair: The experimental evaluation is weak: important baselines are missing, or the results do not adequately support the main claims.

**Q2(5) Reproducibility:**

2: Fair: Key resources (e.g., proofs, code, data) are unavailable but key details (e.g., proof sketches, experimental setup) are sufficiently well-described for an expert to confidently reproduce the main results.

**Q3 Main Strengths:**

The paper is very well written and introduces the problem well. The problem addressed in this paper is very fundamental and can play a useful role in understanding influence or disease propagation in large-scale settings such as social network graphs.

The proposed algorithm is novel and assumes that the set function f_t is submodular in expectation, whereas the prior work in adversarial setting assumes that the set function f_t is submodular.

Algorithm 1 shown in this paper is simple and easy to implement given the reward functions f_t.

The proposed algorithm achieves better performance compared to the chosen baselines DART, CMAB-SM, and OG in most of the cases.


**Q4 Main Weakness:**

Submodular algorithms for MAB problems is really a saturated field and the newer algorithms only have subtle variations to existing methods. While the authors have done an excellent job of showing the different methods in the literature and the bounds in Table 1, it seems incremental and one has to extrapolate carefully as why the proposed extension is interesting. One way to do this is to show applications that can not be dealt with prior methods. Unfortunately, the paper does not clearly convey this.

In contrast to existing methods that assumes f_t to be submodular, the submission assumes that f_t is only submodular in expectation. In order to deal with this generalization, the algorithm 1 computes the empirical mean using m samples where is a very large value since it depends on T. In many of the experiments, T is pretty large and this makes the algorithm computationally very intensive. It would be good to provide the computational complexity of Algorithm 1




**Q5 Detailed Comments To The Authors:**

In all the experiments, it appears that DART starts to outperform ETCG when we increase T. The paper needs to investigate this more and address this issue.

Please see my comments on the main weaknesses.

**Q7 Justification For Your Score:**

While the paper has some weaknesses on the experiments, overall the results are interesting.

**Q9 Complying With Reviewing Instructions:**

1: Yes.

---

### Decision · Program_Chairs · 2022-05-15

**Decision:**

Accept (Poster)

**Comment:**

Meta Review: This paper proposes an explore-then-commit (ETC) algorithm for maximizing submodular functions. This is done under the assumption of a full-bandit feedback, which means that the learning agent observes stochastic values of the maximized functions, but not of the individual arms. The reviewers generally liked the paper and their concerns were addressed by the rebuttal. The rebuttal was competent and one score of the paper improved as a result. My main concern with this paper is novelty. The design of Algorithm 1, phases where arms are added greedily to the already accepted arms, is standard and so would be the analysis.